# UV-Cured Bio-Based Acrylated Soybean Oil Scaffold Reinforced with Bioactive Glasses

**DOI:** 10.3390/polym15204089

**Published:** 2023-10-14

**Authors:** Matteo Bergoglio, Ziba Najmi, Andrea Cochis, Marta Miola, Enrica Vernè, Marco Sangermano

**Affiliations:** 1Dipartimento di Scienza Applicata e Tecnologia, Politecnico di Torino, C.so Duca degli Abruzzi 24, 10129 Torino, Italy; matteo.bergoglio@polito.it (M.B.); marta.miola@polito.it (M.M.); enrica.verne@polito.it (E.V.); 2Department of Health Sciences, Center for Translational Research on Autoimmune and Allergic Diseases—CAAD, Università Del Piemonte Orientale (UPO), 28100 Novara, Italy; ziba.najmi@uniupo.it (Z.N.); andrea.cochis@med.uniupo.it (A.C.)

**Keywords:** bio-based scaffold, 3D printing, bioactive glass, thermosets, photopolymer

## Abstract

In this study, a bio-based acrylate resin derived from soybean oil was used in combination with a reactive diluent, isobornyl acrylate, to synthetize a composite scaffold reinforced with bioactive glass particles. The formulation contained acrylated epoxidized soybean oil (AESO), isobornyl acrylate (IBOA), a photo-initiator (Irgacure 819) and a bioactive glass particle. The resin showed high reactivity towards radical photopolymerisation, and the presence of the bioactive glass did not significantly affect the photocuring process. The 3D-printed samples showed different properties from the mould-polymerised samples. The glass transition temperature T_g_ showed an increase of 3D samples with increasing bioactive glass content, attributed to the layer-by-layer curing process that resulted in improved interaction between the bioactive glass and the polymer matrix. Scanning electron microscope analysis revealed an optimal distribution on bioactive glass within the samples. Compression tests indicated that the 3D-printed sample exhibited higher modulus compared to mould-synthetized samples, proving the enhanced mechanical behaviour of 3D-printed scaffolds. The cytocompatibility and biocompatibility of the samples were evaluated using human bone marrow mesenchymal stem cells (bMSCs). The metabolic activity and attachment of cells on the samples’ surfaces were analysed, and the results demonstrated higher metabolic activity and increased cell attachment on the surfaces containing higher bioactive glass content. The viability of the cells was further confirmed through live/dead staining and reseeding experiments. Overall, this study presents a novel approach for fabricating bioactive glass reinforced scaffolds using 3D printing technology, offering potential applications in tissue engineering.

## 1. Introduction

Bone injuries and disease-related defects pose a significant threat to individuals’ health and quality of life. These bone fractures occur more frequently in older adults, who constitute a growing percentage of the population in many countries worldwide [1]. The human bone possesses the capacity to heal itself, but this property is limited and unable to rectify all defects. In cases where natural healing is insufficient, various techniques have been developed over the years, including bone grafting, bone implants and bone allograft [2]. Tissue engineering, first used in 1987, has gained significant importance and prominence in the field of bone repair and regeneration. This approach combines biocompatible biomaterials that provide bioactive support to cells, mimicking the natural cellular matrix [3].

Larry Hench, who introduced bioactive glasses (BGs) in the research field for the first time, provided the 45S5 Bioglass^®^, the first example of an implant material that is both bioactive and biocompatible [4,5]. BGs’ properties make these materials ideal for replacing fractured or damaged bone. The key feature of BGs lies in their ability to release beneficial ions such as Si, Na, Ca, and P when dissolved in the physiological environment. These ions have the potential to activate and regulate osteogenic genes, promoting bone healing and regeneration in a natural physiological atmosphere [6].

Despite the numerous merits of BGs, their application in scaffolds is constrained due to certain limitations. An ideal scaffold needs to be bioactive, biocompatible and osteoconductive and exhibit excellent mechanical behavior [3]. Unfortunately, BG-based scaffolds often fall short in terms of mechanical characteristics. During the sintering process, it becomes challenging to eliminate voids and pores present in the powder, resulting in a material with low compression and tensile strength, highlighting the need for alternative approaches to overcome this issue in BG-based scaffolds [7].

One approach to enhance the mechanical properties of bioactive glass-containing scaffolds is to incorporate them into a polymer matrix. In cases where a simple surgical process is not sufficient to heal and repair bone defects, hence when the fractures are larger than 2.5 cm, polymeric-BG scaffolds can be employed [8,9]. Polymer-based scaffolds promote tissue growth through optimal mechanical properties, similar to bone properties, and various reinforcements that can enhance adhesion, osteogenic differentiation and cellular proliferation [10].

Natural polymers such as chitosan, collagen, alginate, hyaluronic acid and gelatin have been widely utilised as scaffold matrices due to their biocompatible and biodegradable nature [11,12,13,14,15,16,17,18,19]. In addition, a large number of synthetic polymers have further been evaluated as appropriate scaffolds for the extracellular matrix to mimic bone tissue, including PLGA (poly(lactic-co-glycolic acid), PLA (poly(lactic acid), poly(caprolactone) and poly(vinyl alcohol) (PVA) and other synthetic polymers [10,20,21,22,23]. Both natural and synthetic polymers have their limitations. Natural polymers often exhibit poor mechanical properties [24,25,26], while synthetic biopolymers may have inadequate biocompatibility and degradation reaction and often they rely on unsustainable petroleum-based sources for raw materials [7]. However, recent advancements in the chemical modification of natural biopolymers offer a promising solution for tissue engineering applications. One example is acrylate epoxidized soybean oil (AESO), which consists of a triglyceride structure with acrylate functional groups that can be exploited for the UV-curing process [27,28,29]. These triglycerides are composed of natural fatty acids, do not show cytotoxicity and are easily degraded by human metabolism. Consequently, AESO has gained attention as a viable polymer matrix for scaffold applications in the biomedical field [30,31].

The challenge in using polymers to create scaffolds consists in synthesising the artefact in the desired form, which, considering each patient’s unique requirements, changes in every case. However, with additive manufacturing, the issue can be overcome. Complex three-dimensional architectures can be generated, which can be adapted on a case-by-case basis [32,33]. Among the various 3D printing techniques, vat photo-polymerization is a promising technology for advanced applications since it offers a high surface finish and the highest printing resolution [34,35]. Digital light processing (DLP) 3D printing is included in the vat photopolymerization category, and it offers resolutions ranging from 15 to 100 μm, enabling the construction of complex and hollow parts. DLP uses a digital micromirror device (DMD) to control and redirect UV light with λ between 380 and 405 nm in a precise manner to create a pattern. The pattern, representing a 2D slice of the computer-aided design (CAD) model, is focused onto a photoactive resin, and the layer-by-layer process is repeated while vertically moving the platform to create the final 3D form [36,37,38,39].

If polymeric materials are used, it is also important to carefully analyse their environmental impact and sustainability. It would be very interesting to create a scaffold that possesses the fundamental characteristics of a biocompatible material but also has sustainability. Our research group has deeply investigated the synthesis of sustainable polymers, particularly focusing on photochemistry as an alternative to overcome the energy-consuming nature of traditional synthesis methods [40,41,42,43]. The UV-curing process is more energy efficient than traditional thermal curing, utilising UV-irradiation that requires less energy and can complete the reaction in a few minutes instead of hours. Photo-induced processes can be divided into two categories based on their mechanism: radical and cationic [44,45,46]. Radical photo-polymerisation is particularly effective in reacting with acrylate monomers in inert conditions, as demonstrated in many articles [31,47,48].

This study introduces a novel 3D-printed composite scaffold fabricated through radical photo-polymerisation. The scaffold matrix is composed of a mixture of AESO as polymeric precursor in the presence of different ratios of isobornyl acrylate (IBOA), which is the reactive diluent used to adjust the viscosity of the photocurable formulations to make them printable. The reinforcement is achieved by incorporating synthesised bioactive glass particles, which act as a reinforcing agent and enhance the final product’s biocompatibility.

## 2. Materials and Methods

### 2.1. Materials

Acrylated soybean oil (AESO) and isobornyl acrylate was purchased from Sigma-Aldrich, Milano, Italy. Phenylbis (2,4,6-trimethylbenzoyl) phosphine oxide (Irgacure 819) was purchased from BASF; ammonium hydroxide (NH_4_OH), tetraethyl orthosilicate (TEOS), triethyl phosphate (TEP) and calcium nitrate tetrahydrate (Ca(NO_3_)_2_·4H_2_O) were purchased from Sigma-Aldrich, Milano, Italy.

### 2.2. Bioactive Glass Synthesis

For this work, silica-based bioactive glass particles (named from now on BG S4) have been synthetized, following the modified Stöber method [49], optimized in our previous research [50,51].

The nominal composition of the BG S4 is reported in Table 1.

The synthesis was made by mixing through stirring two different solutions, one made by bi-distilled water, ethanol and ammonium hydroxide (NH_4_OH) and another one made by ethanol and tetraethyl orthosilicate (TEOS). This first step permitted us to create the silica (SiO_2_) particles. Subsequently, the Triethyl phosphate (TEP), phosphor (P) precursor, calcium nitrate tetrahydrate (Ca(NO_3_)_2_·4H_2_O) and calcium (Ca) precursor were added. After the precursors incorporation, the system was thermally treated for 48 h in the oven at 60 °C to remove the residual water, then moved to a furnace for 2 h at 700 °C, with a heating ramp of 5 °C/min to remove all the organic compounds, obtaining spherical BG S4 particles.

### 2.3. Formulation and Photo-Curing

AESO bio-based resin was mixed with 1 wt% of radical photo-initiator Irgacure 819 and a variable amount of reactive diluent and BG S4. The four components, acrylated soybean oil, reactive diluent, BG S4 particles and photo-initiator, were mixed through an ultra-turrax T 10 basic until all the components were homogeneously mixed. We prepared 15 formulations, and their composition is summarized in Table 2. The formulations were then stored in a dark environment to prevent premature curing caused by light contact. After the storage period, the formulations were UV-cured in silicon moulds using a DYMAX ECE flood lamp (Dymax Europe GmbH, Wiesbaden Germany) at a light intensity of 130 mW/cm^2^ for 60 s. Alternatively, they were 3D printed using a Prusa SL1S SPEED (Prusa Research, Prague, Czech Republic), and subsequently post-cured for 60 s under the DYMAX ECE flood lamp.

### 2.4. Characterization

#### 2.4.1. Attenuated Total Reflectance Fourier Transform Infrared Spectroscopy (ATR-FTIR)

The curing process was monitored using a Nicolet iS 50 Spectrometer (Thermo Scientific, Milan, Italy). To analyse the reaction, the viscous resin was spread using a stir bar over a silicon slice with a 32 µm thickness. The spectral resolution of the spectra was 4 cm^−1^, and the data were processed using OMNIC software from Thermo Fisher Scientific.

Double-bond conversion was evaluated by following the decrease of the acrylate peak centred around 1620 cm^−1^, while the peak at 2930 cm^−1^ was taken as reference. Equation (1) was utilised to evaluate the conversion during irradiation.
(1)$$Conversion\left(\%\right)=\frac{({\frac{A_{group}}{A_{ref}})}_{t=0}-({\frac{A_{group}}{A_{ref}})}_{t}}{({\frac{A_{group}}{A_{ref}})}_{t=0}}\times 100$$
where A_group_ corresponds to the acrylate group area investigated and A_ref_ is the reference area at 2390 cm^−1^.

#### 2.4.2. Photo Dynamic Scanning Calorimetry (Photo-DSC)

The crosslinking reaction progress was monitored using a photo-DSC instrument. The instrument setup consisted of a Mettler TOLEDO DSC-1 (Milan, Italy) equipped with a Gas Controller GC100 (Milan, Italy) and a mercury lamp, the Hamamatsu LIGHTINGCURE LC8 (Hamamatsu Photonics (Milan, Italy)), which was used with an optic fibre to focus the radiation.

The UV-light employed was settled at 365 nm wavelength, and the intensity was set to 10% of the maximum intensity, resulting in 10 mW/cm^2^. Samples weighing between 5 to 15 mg were placed in an open aluminum pan, while an empty aluminum pan was used as a reference. All the tests were conducted under a nitrogen flow of 40 mL/min at room temperature (25 °C).

The evaluation method involved the settling of the sample for two minutes, followed by exposure to UV-light for two irradiation steps of ten minutes. The second UV-light exposure was necessary to ensure complete conversion of the groups and to create the baseline. In fact, the second curve was subtracted from the first one to obtain specifically the curve related to the reticulation.

All the data were elaborated by Mettler Toledo STARe software V9.2.

#### 2.4.3. Rheology and Photo-Rheology

Rheology and photo-rheology of the thermoset precursor resin were performed using the Anton Paar MCR302 (Turin, Italy) parallel plate rheometer. Rheology was conducted to evaluate the viscosity of the laboratory-prepared formulations, determining their suitability for the 3D printing process. For this analysis, two plates with a diameter of 25 mm were used, with a 1 mm gap between them. Viscosity values were recorded over a shear rate range of 0.01 to 1000 s^−1^.

Photorheology was also conducted using the same rheometer, with the addition of a Hamamatsu LC8 UV lamp with an irradiation power of 30 mW/cm^2^. The lamp was turned on 60 s after the start of each test. However, for this analysis, the lamp was operated at 50% of its intensity, resulting in an irradiation power of 15 mW/cm^2^. The lower plate was replaced with a glass plate to allow the passage of UV radiation, and the gap between the plates was reduced to 0.3 mm. Measurements were performed at a constant frequency of 1 Hz and a constant temperature of 25 °C. During this type of analysis, the two moduli G′ and G″ were measured over time. G′ represents the storage modulus, which describes the elastic component of the material during deformation, while G″ corresponds to the dissipative modulus that represents the viscous component.

#### 2.4.4. Dynamic Mechanical Thermal Analysis (DMTA)

The UV-cured materials were analyzed under a dynamic mechanical thermal analysis, conducted using a Triton Technology device. The analysis began at 0 °C, and the measure was stopped at 100 °C, with a heating rate of 5 °C/min. Uniaxial tensile stress was applied by the device at a frequency of 1 Hz. The main objective of the analysis was to determine the glass transition temperature, which corresponds to the peak of the tan δ curve. The final temperature of the test was selected after the material’s rubbery plateau region. The samples used for the analysis had dimensions averaging 1.5 × 3.5 × 12 mm, and they were either obtained through a silicon mould and UV-cured using a DYMAX ECE flood lamp (Dymax Europe GmbH, Wiesbaden, Germany) at a light intensity of 130 mW/cm^2^ or 3D printed with the Prusa SL1S SPEED (Prague, Czech Republic).

The number of crosslinks per volume ν_c_ was obtained using Equation (2).
(2)$$v_{c}=\frac{E^{\prime }}{3RT}$$
where E′ corresponds to the storage modulus in the rubbery plateau (T_g_ + 50 °C), T is the temperature where E′ is taken in Kelvin and R is the gas constant.

#### 2.4.5. 3D Printing Process

The 3D printing process was carried out with the commercial Masked Stereolitography Apparatus (MSLA) printer (SL1S SPEED purchased from Prusa, Czech Republic). The printer was equipped with a monocromatic 405 nm 25 W UV LED source. The printing was followed by a post curing in a DYMAX lamp (Dymax Europe GmbH, Wiesbaden, Germany) for 60 s to complete the photo-crosslinking.

#### 2.4.6. Compression Test

Scaffold mechanical compression properties were determined through the evaluation of stress–strain curves obtained using a compression instrument (3220 Base System, Electroforce^®^, TA Instruments, New Castle, DE, USA). The translation speed was set to 1 mm/min, and the dimension of the samples had an average of 10 × 10 × 4 mm^3^, following ISO 604:2002 standards [52]. The compressive modulus E_c_ was determined in the linear region of the curve through Equation (3), where E_c_ is the compressive modulus expressed in MPa, σ is the stress in MPa and ɛ is the nominal strain expressed as a dimensionless ratio. All the results derived from an average of 5 samples.
(3)$$E_{c}=\frac{\sigma_{2}-\sigma_{1}}{\varepsilon_{2}-\varepsilon_{1}}$$

#### 2.4.7. Composite Scaffolds Characterization

A comprehensive characterization of composite 3D-printed scaffolds was conducted, focusing on morphology, composition, in vitro reactivity in simulated body fluid (SBF) following Kokubo’s protocol [53], cytocompatibility and metabolic activity. The composite scaffolds were analyzed using field emission scanning electron microscopy (FESEM) with energy dispersive X-ray spectroscopy (EDS) capabilities (SUPRATM 40, Zeiss, Oberkochen, Germany), to assess their morphological and compositional properties. Specimens obtained by breaking DMTA samples into a brittle fracture were fixed to aluminum (Al) stubs using a silver-based adhesive, metallized with platinum (Pt), and examined.

### 2.5. Cytocompatibility and Metabolic Activity Evaluation

#### 2.5.1. Cells Cultivation

Human bone-marrow mesenchymal stem cells (bMSC) were purchased from PromoCell (C-12974) and cultivated in low glucose Dulbecco’s modified Eagle medium (DMEM; Sigma-Aldrich, Milan, Italy) supplemented with 10% fetal bovine serum (FBS; Sigma-Aldrich, Milan, Italy) and 1% antibiotics at 37 °C, 5% CO_2_ atmosphere. Cells were cultivated until 80–90% confluence, detached by a trypsin EDTA solution (0.25% in PBS), harvested and used for experiments.

#### 2.5.2. Cytocompatibility Evaluation

Cells were directly seeded onto specimens’ surfaces at a defined density (2 × 10^4^ cells/sample), and after 4 h of allowing adhesion, 500 μL of culture media was added to each sample. Subsequently, they were cultivated for 24 and 48 h; at each time point, the viability of the cells was evaluated using metabolic activity using the resazurin metabolic assay alamar blue (alamarBlue™, ready-to-use solution from Life Technologies, Milan, Italy) by directly adding the dye solution (0.015% in phosphate buffer saline (PBS)) onto the infected specimens; after 4 h incubation in the dark, the fluorescent signals (expressed as relative fluorescent units (RFU)) were detected at wavelength 570 nm and 590 nm for excitation and emission reading, respectively, by spectrophotometer (Spark, from Tecan, Switzerland); moreover, the fluorescent live/dead assay was applied to visually check for viable cells (LIVE/DEAD, Viability/ Cytotoxicity Kit for mammalian cells, Invitrogen, Milan, Italy) with a digital EVOS FLoid microscope (from Life Technologies, Milan, Italy). Finally, scanning electron microscopy (SEM; JSM-IT500, JEOL, Japan) imaging was used to evaluate surface-attached cells’ morphology; briefly, specimens were dehydrated by the alcohol scale (70–90–100% ethanol, 1 h each), dried with hexamethyldisilazane, mounted onto stubs with conductive carbon tape and covered with a gold layer. Images were collected at different magnifications using secondary electrons.

## 3. Results and Discussion

### 3.1. Photo Curing Process

The UV-curing process of AESO-based formulations was deeply investigated using three different methods, ATR-FTIR, photo-DSC and photo-rheology.

ATR-FTIR analysis was performed to evaluate the acrylate double bond conversion upon UV-irradiation following the decrease of the peak centered at around 1620 cm^−1^.

The conversion curves as a function of irradiation time are reported in Figure 1 for the pristine formulation C0 and for the same formulation containing increasing bio-glasses content. This formulation is taken as an example since, as will be discussed below, it will be the printable formulation investigated in the following part of this work. The conversion data for all the investigated formulations are collected in Table 3. The other graphs obtained by ATR-FTIR are reported in Appendix A.

From the results reported in Table 3, it is possible to observe the very high double bond conversion achieved by all the pristine AESO-IBOA formulations. By increasing the IBOA content in the formulation, there was a slight enhancement of the conversion upon irradiation. This could be attributed to a delay of vitrification, induced by the decrease of the crosslinking density by increasing the IBOA content, which allows a higher double bond conversion. The addition of the bio-glass induced a slight decrease of the overall final double bond conversion but without a significant effect. This could be attributed to a competitive absorption effect of light between the glasses and the photoinitiator, with a decrease of photoinduced reactive initiating species.

Photo-DSC experiments were conducted to validate the ATR-FTIR analysis. The hexotermicity data for all the investigated formulations are presented in Table 4. In Figure 2, an example for the same C formulation with different bio-glass content is reported.

The data reported in Table 4 agree with the FTIR analysis. It is possible to observe a slight increase of exothermicity for the pristine UV-curable formulation by increasing the IBOA content, as observed from the FTIR data. As well, analyzing the photo-DSC data, it is possible to observe a reduced integral area increasing BG S4 content. This behavior is consistent with FTIR analysis. However, it is important to note that the overall reduction in the entire process is negligible and does not significantly impact the desired outcome.

The formulation was also subjected to photo-rheology analysis to investigate the optimal conditions for a 3D printing process and further confirm the curing process. In Figure 3a,b all the rheological curves upon irradiation are reported. It is possible to observe that all the formulation reaches a constant G’ modulus simultaneously (20 s), meaning there is no difference when adjusting both IBOA and BG S4 content in the photocurable formulations. Furthermore, the gelation time, which was approximately 5 s, remained constant varying the composition. These findings provide further evidence about the curing process, confirming the data obtained by FTIR and photo-DSC analysis, and give information about the 3D-printing parameters.

Based on these results, it can be concluded that the photocurable formulations exhibit a favorable conversion rate even varying BG S4 content up to 30 wt%. These characterization methods demonstrate that the incorporation in the formulation does not significantly hinder the overall conversion process.

### 3.2. Thermal and Mechanical Properties of Cured AESO-Based Scaffolds

A complete visco-elastic characterization of the photocured AESO-based materials was obtained by dynamic-mechanical thermal analysis (DMTA). All the data are collected in Table 5. For the pristine formulations, it is possible to observe a slight increase of T_g_ by increasing the IBOA content. This could be due both to the delay on vitrification with an enhancement of the final conversion as well as to the IBOA structure itself. The ν_c_ values shown in Table 5 demonstrate that increasing IBOA content leads to a decrease of ν_c._ This is due to the addition of the monofunctional reactive diluent. Therefore, we would have expected a lower T_g_ of the crosslinked material [54], but we observe an opposite trend with an enhancement of T_g_ increasing the IBOA content. This can be attributed to the chemical structure of isobornyl acrylate owning a rigid ring, which imparts higher rigidity to the polymer network structure notwithstanding the lower crosslinking density. Therefore, a higher content of IBOA leads to increased rigidity in the final structure and consequently an increase of T_g_, even if the crosslinking density ν_c_ decrease. This means that the contribution to the stiffening of the UV-cured polymer network is primarily governed by the chemical structure of the IBOA monomer.

The addition of the BG S4 induced a slight decrease of the final T_g_ compared with the same pristine formulation. The T_g_ decrease observed with the increase of BG S4 content can be attributed to a partial hindering of the overall crosslinking process as previously discussed, due to the competitive light adsorption effect.

The properties of the 3D-printed samples exhibit a different behaviour compared to the samples crosslinked in the mould. Interestingly, the T_g_ shows an opposite trend in the 3D-printed samples with an enhancement of T_g_ by increasing BG S4 content in the photocurable formulation. This can be explained taking into consideration the different crosslinking process.

In 3D printing, the curing is made layer by layer, with each layer of 50 µm polymerised individually. On the other hand, in the mould-polymerised sample, the entire 150 µm thickness is polymerised simultaneously. In the 150 µm thickness formulation, there are more BG particles that hinder the UV-light penetration throughout the sample.

In addition, the 3D printing process requires the post curing in a Dymax lamp for 1 min, which leads to higher conversion of unreacted acrylates groups. The higher T_g_ is then attributed to the improved interaction between the BG S4 and the polymer matrix in the 3D-printed samples, as a consequence of the layer-by-layer curing process.

### 3.3. Rheology

Rheological analysis was performed to investigate the viscosity and dynamic behaviour of the formulations. Figure 4a shows the viscosity curves as a function of shear rate applied for all the investigated formulations, while Figure 4b shows the viscosity for the formulation without IBOA as a reference (named E).

Table 6 provides viscosity values at a shear rate of 30 s^−1^, typical of a 3D printing process [55]. According to the literature [55,56,57,58,59,60,61], the optimal formulation for a 3D printing process is found to be the one named C30, composed by 70:30 AESO to IBOA ratio and 30 phr BG S4. This formulation was then chosen for the consecutive 3D printing process.

### 3.4. 3D Printing Process

The pristine formulation A7I3 and the same formulation containing BG S4 in the ratio between 10 and 30 phr (C10 and C30, respectively) were successfully printed using the PRUSA SL1S printer (Prague, Czech Republic). The printing process was made layer by layer of 50 µm, irradiated under UV-light of 405 nm for 2.5 s. After printing, each sample was immersed in isopropanol and placed inside a sonication bath for 5 min to remove any residual uncured resin. Finally, the samples were placed inside the DYMAX lamp for 1 min to complete the photo-crosslinking (post-curing phase). Various shapes, including DMTA, tensile and compression samples, were printed. An example of a printed structure achieved using the formulation C30 is shown in Figure 5. A complex porous structure with small pores of approximately 1 mm was successfully achieved in the 3D samples.

SEM analysis of the fracture surfaces of 3D-printed samples revealed an optimal distribution of BG S4 within the polymeric matrix. Figure 6 reports the SEM images relative to samples containing 10 and 30 phr BG S4. The images show that the bioactive glass particles were well dispersed throughout the samples, and the aggregates were limited in amount and dimension. Figure 7 shows the presence of single dispersed particles with an average dimension of 489 ± 54 nm.

### 3.5. Compression Tests

Compression tests were performed on the optimised formulation for the 3D printing that corresponded to the formulation C, pristine and charged with 10 and 30 phr BG S4. From the compression tests was determined the compression modulus in the linear region, obtained through Equation (3). The results in Table 7 and Figure 8 demonstrate a decrease of the modulus with increasing bioactive glass content.

Considering the mould-printed samples, the observed decrease of the modulus can be attributed to the lower T_g_ and lower conversion of samples with higher BG S4 content, as previously discussed. These factors contribute to a reduction in the compression resistance of the material. Furthermore, a notable increase of the modulus is observed in the 3D-printed samples compared to the mould-printed samples, indicating that the 3D printing method enhances the overall mechanical characteristics of the final product, as already pointed out.

Focusing on the 3D-printed samples, it is still evident that the modulus decreases with increasing BG S4 content. However, the formulation C30 exhibits higher compressive modulus than C10. This can be attributed to two factors: the addition of BG S4 initially reduces the mechanical properties of the sample due to hindered photopolymerization. However, during the 3D printing process, the incorporation of BG S4 is improved, resulting in an increase of mechanical properties. Thus, beyond 30 phr, there is a synergistic effect between the composite and the reinforcement provided by the BG S4, contributing to an increase in material resistance.

### 3.6. Cytocompatibility Evaluation

According to the results obtained from the above experiments, such as the rheology test and number of crosslinks per volume, the AESO:IBOA (70:30) formulation with 30BGs (C30) had potential properties in terms of being used for 3D printing. To investigate its cytocompatibility effect on human cells, human bone-marrow mesenchymal stem cells were selected as a candidate cell line due to their self-renewal and differentiation abilities that play a pivotal role in tissue healing and regeneration [62]. For evaluation of whether the cytocompatibility activity of the samples would be changed by a modification in the number of BGs or IBOA, the following samples, AESO: IBOA (70:30) with 0 and 10 BG (C0 and C10, respectively), AESO: IBOA (50:50) with 0 BG (A0) and E0 with 100% of AESO without BG and IBOA, were selected to compare their cytocompatibility evaluations in the in vitro condition; only hMSC cells seeded in the 24-multiwell plate without any samples were considered control samples (their results are shown in Appendix A). Briefly, hMSCs were directly seeded on the samples’ surfaces, and after 24–48 h, the viability and morphology of attached and spread cells were analyzed by metabolic assay resazurin (alamar blue), fluorescent live/dead stain and SEM images, respectively, as reported in Figure 9a,b and Appendix A. Twenty-four hours after seeding the cells and incubation, the metabolic activity of hMSCs adhering to the surfaces of A0, C0 and C10 was reduced to 60, 70 and 57%, respectively. Since the metabolic activity, which is defined as the RFU value, of the cells attached to the surfaces of the E0 samples was statistically similar to that of the only hMSCs without any samples (*p* > 0.05, Appendix A), the E0 specimens were considered control samples with 100% metabolic activity, and the obtained results were normalized with that. The high metabolic activity of the cells on the C30 samples’ surfaces indicated that a high number of the alive cells attached to their surfaces metabolized a non-fluorescent and blue resazurin into a fluorescent and pink component named resorufin, and their intensity was measured at an emission wavelength of 590 nm. The same results of the metabolic activity were observed after 48 h, demonstrating that C30 samples were cytocompatible with hMSCs, and statistically significant differences were observed between the cells attached to their surfaces and other tested samples (A0, C0, C10 and E0; Figure 9a, *p* < 0.01 indicated by **). SEM images were taken 48 h after seeding cells confirmed the results of metabolic activity indicating that C30 with the formulation of AESO:IBOA (70:30) with 30BG created a suitable surface for cells’ attachment and proliferation, while on the surfaces with a lower ratio of AESO: IBOA (A0, E0) and a low number of BGs (C0 and C10), a few cells attached and some non-adhered cells (round-shaped) were observed (Figure 9b). The viability of surface-attached cells was investigated by fluorescent live/dead staining, and the results are presented in Appendix A. A high number of alive hMSC cells, colored green, attached and spread on the surface of the C30 samples in comparison with the other samples with low phr BG (A0, C0, C10 and E0; Appendix A). To confirm whether the round-shaped cells detected in SEM images were alive or dead, 24 h after seeding the hMSCs on the specimens’ surface, surplus culture medium was collected and added to a new 24-multiwell plate. The aim was to reseed floating cells in the new multiwell plate, and then their attachment and growth were visually monitored; the results are shown in Figure 9c. As was shown in the control samples (E0) that contained 100% AESO with no IBOA and BG, about half the number of cells are attached to the bottom surface of the well, and by increasing the ratio of AESO:IBOA up to 50:50 and 70:30, the number of alive cells (attached ones) decreased with respect to the control samples; even an increase of the amount of BG up to 10 phr showed no changes in the cells’ viability. However, the same as the results obtained from metabolic activity and SEM images, few dead cells were observed for C30 after reseeding the floating cells in the new multiwell plate (Figure 9c).

The results of the cytocompatibility evaluation agreed with the above-obtained results, such as rheology and dynamic mechanical thermal analysis, confirming that C30 samples containing AESO:IBOA (70:30) formulation with 30 BG were not only suitable for 3D printing but also showed a promising result for compatibility on human cells. As the previous literature showed, polymeric matrixes coupled with bioactive glass improve their mechanical performance as scaffolds and increase their properties from a biological point of view, such as cell growth and proliferation, simultaneously [63]. This improvement can be due to an alkalinising ability by releasing some ions (Na^+^, K^+^ and Ca^2+^) and neutralization of the acidic environment caused by degradation of some polymers [64]; additionally, BG S4 establish favourable sites for cells attachment and proliferation and play a stimulatory role for bone formation by releasing Si ions [65].

## 4. Conclusions

A bio-based acrylated resin derived from soybean oil was combined with a reactive diluent, isobornyl acrylate, to synthetize a bioactive glass (BG S4) reinforced scaffold. The formulation consisted of four components: acrylated epoxidized soybean oil (AESO), isobornyl acrylate (IBOA), the photo-initiator Irgacure 819 and the silica-based BG S4. In addition to using a bio-based precursor known as biocompatible [31], a reactive diluent was used to adjust the density of the formulation without affecting biocompatibility. The starting polymeric formulation was added with a 10 and 30 phr amount of BG S4 to enhance biocompatibility.

The resin exhibited high reactivity towards radical photopolymerisation, and the presence of BG S4 had minimal impact on the photocuring process. The UV-cured BG S4 reinforced networks were characterised in terms of viscoelastic and thermal-mechanical properties by means of DMTA and compressive tests. Samples containing increasing amounts of BG S4 showed slightly lower glass transition temperature but higher crosslinking density.

Rheology studies performed on the uncured resin indicated that the formulation with a 70:30 ratio of AESO to IBOA added with 30 phr BG S4 (named C30) was optimal for the MSLA (masked stereolitography) 3D printing process. The mechanical characterisation of the 3D-printed sample revealed a compressive elastic modulus of about 30 MPa, which was higher than the mould UV-cured samples, especially comparing the formulation containing 30 phr of BG S4, suggesting the high efficacy of the 3D printing process in producing mechanically stable scaffolds.

Furthermore, in vitro cytocompatibility evaluation of the UV-cured samples with and without bio-glass towards bMSCs demonstrated a considerable increase of metabolic activity of cells on the C30 surfaces; according to visual observation by means of fluorescent live/dead and SEM images, it can be due to a high number of attached cells to their surfaces.

## Figures and Tables

**Figure 1 polymers-15-04089-f001:**
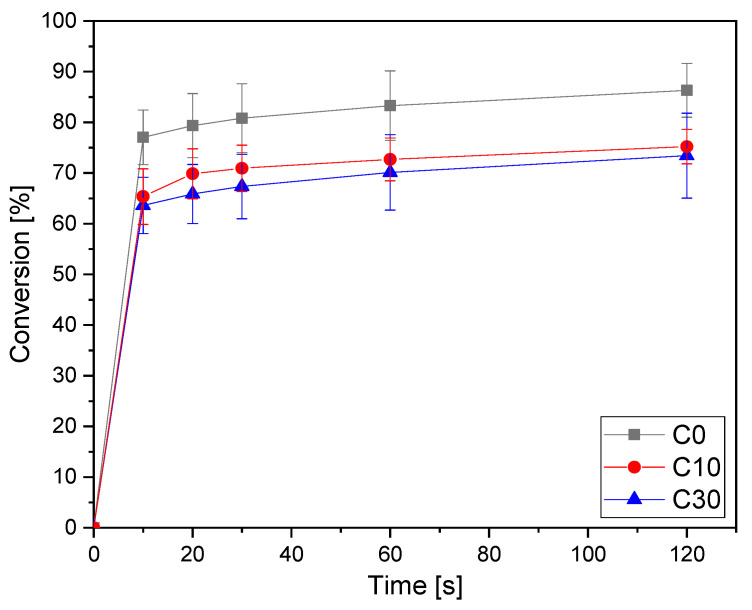
Conversion curves as a function of irradiation time for the formulation C (from ATR-FTIR) as a function of time varying bio-glass concentration. Light intensity was set at 130 W/cm^2^.

**Figure 2 polymers-15-04089-f002:**
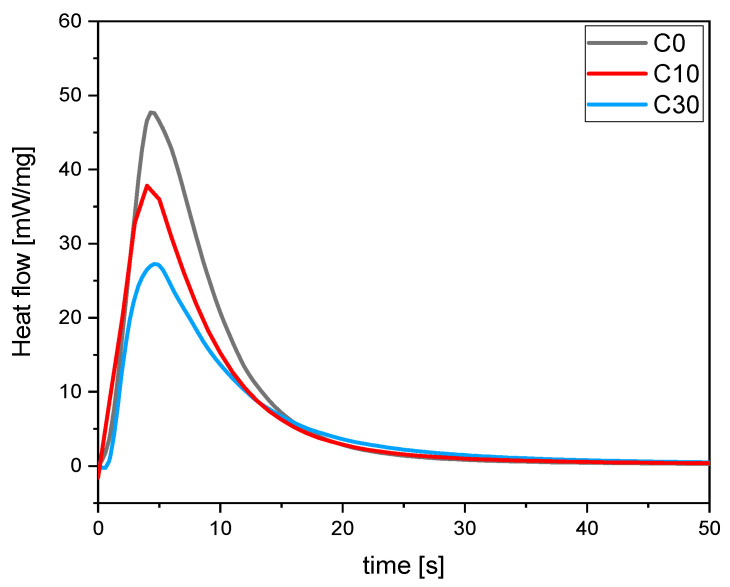
Heat released during UV-light irradiation of C formulation as a function of time changing bio-glasses concentration.

**Figure 3 polymers-15-04089-f003:**
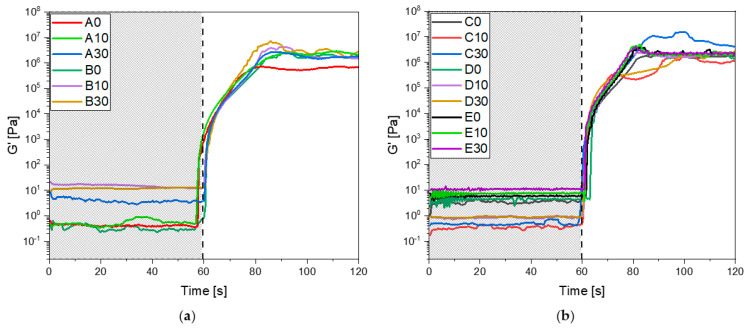
(**a**) Photo-rheology results of formulation from A to B; (**b**) photo-rheology results of formulation from C to E. For all the tests the UV-light irradiation started after 60 s.

**Figure 4 polymers-15-04089-f004:**
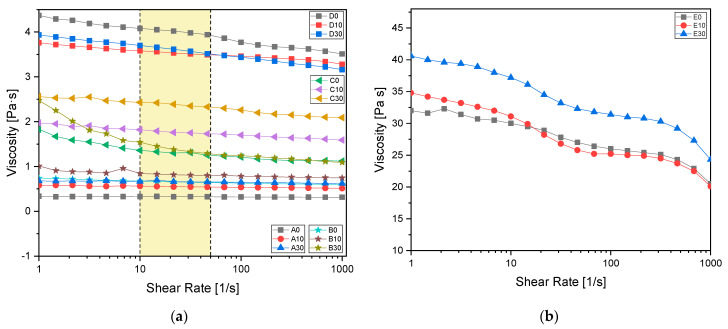
(**a**) Formulation viscosity measured from 1 to 1000 shear rate [1/s] with a plate of 2.5 mm diameter. In yellow the shear rates typical of a 3D printing process are evidenced; (**b**) viscosity reference of AESO samples without reactive diluent. All the values were measured with a 2.5 mm diameter plate.

**Figure 5 polymers-15-04089-f005:**
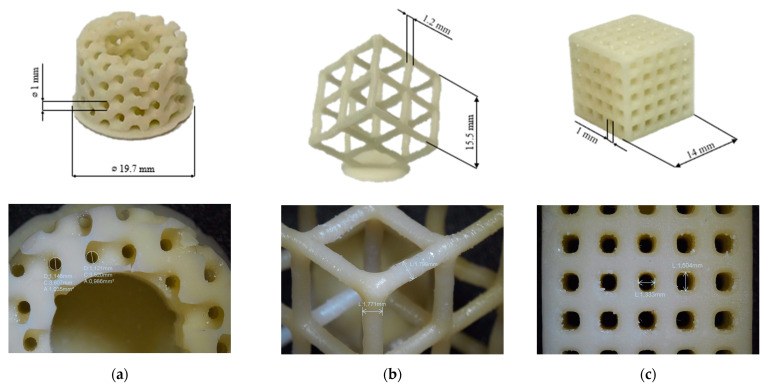
(**a**–**c**) Represents different porous 3D-printed structures with formulation C30 with their magnification.

**Figure 6 polymers-15-04089-f006:**
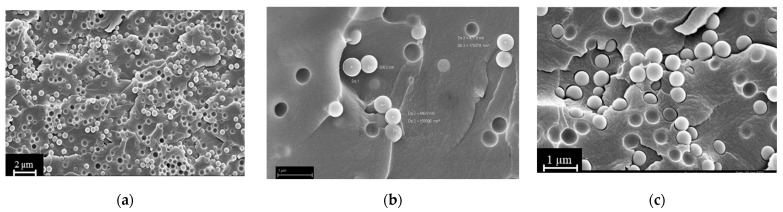
(**a**) Sample C30 at 1000× magnification; (**b**) sample C10 at 30,000× magnification; (**c**) sample C30 at 30,000× magnification.

**Figure 7 polymers-15-04089-f007:**
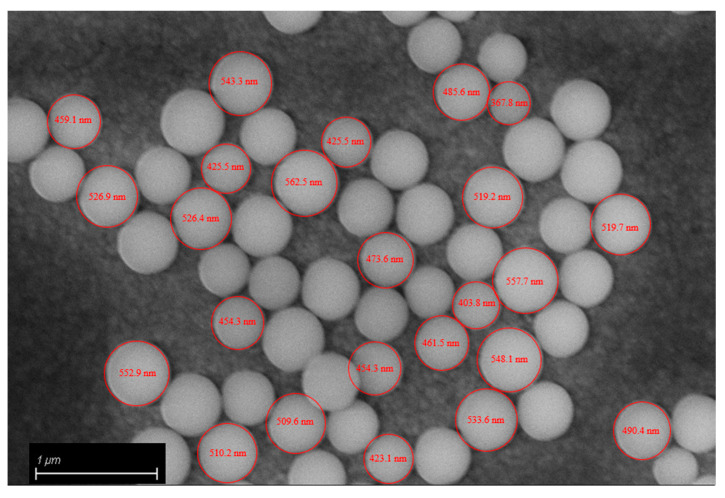
Bioactive glass diameter.

**Figure 8 polymers-15-04089-f008:**
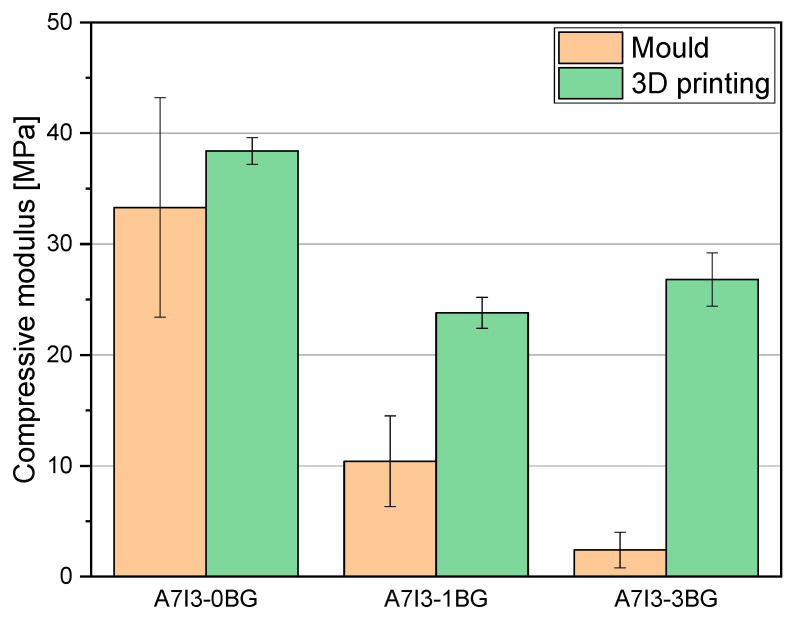
Histogram representation of compressive modulus of C samples mould printed and 3D printed.

**Figure 9 polymers-15-04089-f009:**
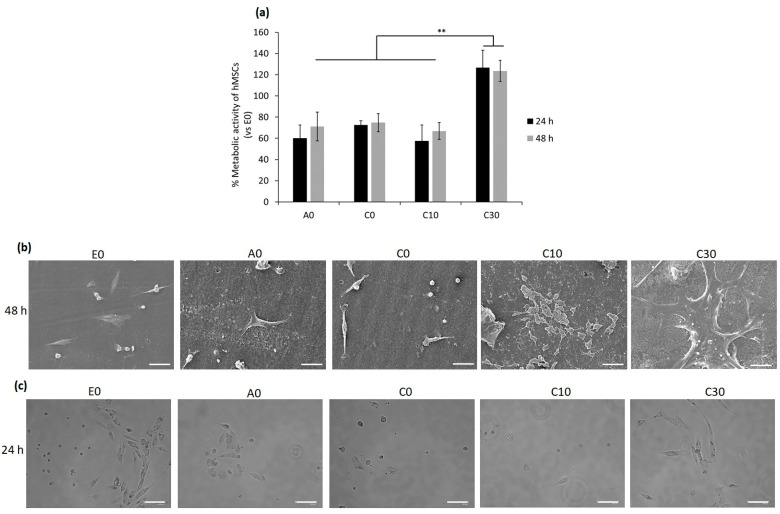
Cytocompatibility evaluation of the specimens with different ratios AESO:IBOA and phr BGs towards bMSCs. **Direct seeding cells on to the specimens’ surfaces:** (**a**) Metabolic activity of the cells after 24 and 48 h; the results were normalized with E0 as control samples (** indicates *p* ˂ 0.01); (**b**) SEM images of the surface-attached cells after 48 h (scale bar is 50 µm); reseeding the floating cells; (**c**) bright field microscope images, 24 h after reseeding the floating cells in a new 24-multiwell plate (scale bar = 100 µm).

**Table 1 polymers-15-04089-t001:** Nominal composition of bioactive glass S4.

Sample Name	Composition %wt		
BG S4	SiO_2_	P_2_O_5_	CaO
	77	9	14

**Table 2 polymers-15-04089-t002:** Formulation AESO:IBOA with a variable amount of BG S4 particles.

AESO (%wt)	IBOA (%wt)	BG S4 (phr)	Sample Name
50	50	0	A0
		10	A10
		30	A30
60	40	0	B0
		10	B10
		30	B30
70	30	0	C0
		10	C10
		30	C30
80	20	0	D0
		10	D10
		30	D30
100	0	0	E0
		10	E10
		30	E30

**Table 3 polymers-15-04089-t003:** Final conversion after 120 s UV-light irradiation.

Sample Name	Conversion after 120 s Irradiation
A0	82 ± 5
A10	76 ± 2
A30	75 ± 5
B0	80 ± 8
B10	72 ± 2
B30	66 ± 3
C0	86 ± 8
C10	74 ± 2
C30	73 ± 14
D0	77 ± 2
D10	74 ± 4
D30	65 ± 14

**Table 4 polymers-15-04089-t004:** Heat released during photo-DSC analysis with different amounts of IBOA and bioactive glasses.

Sample Name	Integral [J/g]
A0	459 ± 11
A10	418 ± 4
A30	338 ± 5
B0	437 ± 10
B10	351 ± 5
B30	342 ± 7
C0	419 ± 7
C10	350 ± 10
C30	315 ± 2
D0	389 ± 11
D10	316 ± 13
D30	275 ± 5

**Table 5 polymers-15-04089-t005:** Results obtained by DMTA analysis for samples with variable BG S4 and IBOA rates. T_g_ was calculated as the maximum of tan δ. $v_{c}$ calculated by Equation (2).

Sample Name	Glass Transition Temperature T_g_ [10,251]	Number of Crosslinks per Volume ν_c_ [mol/m^3^]
A0	75 ± 1	3408
A10	74 ± 0	6553
A30	73 ± 0	6096
B0	73 ± 2	7734
B10	72 ± 0	7508
B30	68 ± 3	10,251
C0 (3D printed)	67 ± 2 (60 ± 0)	11,394 (6162)
C10 (3D printed)	66 ± 3 (64 ± 4)	189,063 (8492)
C30 (3D printed)	60 ± 0 (72 ± 2)	180,971 (8676)
D0	61 ± 0	12,250
D10	62 ± 3	17,412
D30	60 ± 2	20,726

**Table 6 polymers-15-04089-t006:** Viscosity value measured at 30 s^−1^ with a parallel plate of 2.5 mm diameter.

Sample Name	Viscosity [Pa*s] at 30 s^−1^
A0	0.33
A10	0.55
A30	0.66
B0	0.65
B10	0.81
B30	1.33
C0	1.31
C10	1.75
C30	2.35
D0	3.98
D10	3.51
D30	3.57
E0	27.80
E10	26.80
E30	33.20

**Table 7 polymers-15-04089-t007:** Compressive modulus obtained from compression test.

	Compression Modulus [MPa]		
AESO-IBOA Ratio	0 BG	1 BG	3 BG
C (mould)	33.3 ± 9.9	10.4 ± 4.1	2.4 ± 1.6
C (3D printed)	38.4 ± 1.2	23.8 ± 1.4	26.8 ± 2.4

## Data Availability

The data presented in this study are available on request from the corresponding author.

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
