# Peer review of "UV-Cured Bio-Based Acrylated Soybean Oil Scaffold Reinforced with Bioactive Glasses"

_polymers, 2023, doi:10.3390/polym15204089_

Round 1

Reviewer 1 Report

The manuscript submitted investigated a composite scaffold from bio-based acrylate resin derived from soybean oil in combination with a reactive diluent, isobornyl acrylate, and reinforced with bioactive glass particles. In addition, they introduced a novel 3D-printed composite scaffold fabricated through radical photo-polymerization. Overall, most of the conclusions are supported by experimental evidence. However, the manuscript lacks some essential pieces of information. Major revision is needed. My further comments are given below.

There are so many errors (Error! Reference source not found.) during format conversion which makes the manuscript difficult to read. Although it’s not authors fault but they should double check it.

The figure numbers are confusing. There are two figure 1, figure 2 and figure 3!

The abstract needs to be more attractive by indicating the novelty of the work.

Biological characterization should be removed from keywords.

The BG S4 is not a commercial glass but derives from our previous research? In addition, references are missing in lines 137 and 152 and many other places.

Line 271:By increasing the IBOA content in the formulation there is a slight enhancement of the conversion upon irradiation. This could be atributed to a delay of vitrifcation, induced by the decrease of the crosslinking density by increasing the IBOA content, which allows a higher double bond conversion. How to explain the delay of vitrification? Is there a decline of the crosslinking density by raising the IBOA content?

In figure 2 Heat released during UV-light irradiation of A7I3 formulation as a function of time changing bio-glasses concentration. As I noticed, it increased from 1-8 min, while 9-16 min showed a decreasing trend and later stabilized. Why?

What’s the specific differences in metabolic activity observed between the A7I3-3BG formulation and other samples after 24 hours?

How does increasing the content of isobornyl acrylate (IBOA) affect the glass transition temperature (Tg) in pristine formulations?

Acceptable

Author Response

reply to the reviewer in the attached file

Reviewer 2 Report

Authors present a research article focused on the design, fabrication and characterization of either 3D printed or mold casting scaffolds based on acrylated soybean oil and bioactive glasses. Authors include in the fabrication process the use of UV to produce the scaffolds. Overall, this study presents a well-structured manuscript introducing an interesting approach for fabricating novel biomaterials offering potential applications in tissue engineering, and can be helpful for a broad audience interest into developing novel materials for bio applications.

My remarks:

1.- After reading the paper and the bibliography, I’ve found 6 self-citation over 68. All these works related with the topic. Also, authors include an up-to-date bibliographic work including several 2023 relevant works on the field.

2.- Through the text, authors must correct the writing of different chemical such as SiO2 must be SiO2 (line124) etc the under scrip is missing

3.- line 138 reference is missing

4.- Could authors mention if they used fresh prepared resin each time? In view of his reactivity to UV if a resin was stored for 1 day or 1 month can this alter the samples?

5.- Could author also mention which one of the two process they consider the most efficient? The use of silicon molds or the use of 3D printers?

6.- very long nomenclature for label the samples. Hard to follow

7.- some information could be moved to SI such as table 3 …

8.- Figure 1 is missing error bars?

9.- Use 1 single significant figure in errors (table 4), max 2 significant figure but please don t use more that 2!!!

10.- authors include the G’ and G’’ from rheology to explain the visco elasticity of the sample. A table regarding the most important parameters from these curves would be nice, complete table 6 with these values. Perhaps, is better to start with viscosity and the G’ and G’’?. I suggest also to separate in two rows the A515 from the A6I14 series. Viscosity curves are often better log-log

11.- line 315, 354 reference missing

12.- What happen with cell control (cells cultured without the presence of the material) ? what is their metabolic activity compared to those cells treated with the materials?

13.- in general figures are blurry, and not justified in the text or aligned.

14.- I also think that the work lacks of an experimental characterization to indicate the presence of components of the composite and there are not depredated after the processing (to demonstrate that at the end your composite has soybean, BG etc.

15.- I ve checked on the WOS and I did not find similar work \addressing the present research

16.- Overall, I do believe that the presented manuscript is interesting for a broad community of researcher and industrial. After major revision, this paper would be suitable to be published.

Author Response

(The authors gave the same response as above.)

Round 2

Reviewer 1 Report

The authors have addressed all my comments and the manuscript has been improved. I have no further comments on the revised manuscript.

It's fine.

Reviewer 2 Report

authors responded reviewer comments accordingly